# Screening the Coulomb interaction leads to a prethermal regime in two-dimensional bad conductors

L. J. Stanley[1,2], Ping V. Lin [1,3], J. Jaroszyński [1] & Dragana Popović [1,2] ✉

The absence of thermalization in certain isolated many-body systems is of great fundamental interest. Many-body localization (MBL) is a widely studied mechanism for thermalization to fail in strongly disordered quantum systems, but it is still not understood precisely how the range of interactions affects the dynamical behavior and the existence of MBL, especially in dimensions $D > 1$. By investigating nonequilibrium dynamics in strongly disordered $D = 2$ electron systems with power-law interactions $\propto 1/r^\alpha$ and poor coupling to a thermal bath, here we observe MBL-like, prethermal dynamics for $\alpha = 3$. In contrast, for $\alpha = 1$, the system thermalizes, although the dynamics is glassy. Our results provide important insights for theory, especially since we obtained them on systems that are much closer to the thermodynamic limit than synthetic quantum systems employed in previous studies of MBL. Thus, our work is a key step towards further studies of ergodicity breaking and quantum entanglement in real materials.

One of the most fundamental questions in statistical mechanics is the state that an isolated quantum many-body system reaches a long time after a perturbation. Is the system able to achieve thermal equilibrium, or does it fail to thermalize and why? According to the ergodic hypothesis, different parts of an isolated system exchange energy and particles and thus the system reaches thermal equilibrium at long times; such systems are then described by statistical mechanics at equilibrium. In recent years, however, there has been a growing interest[1–5] in mechanisms that break ergodicity, in particular, in many-body localization (MBL)[6,7]. When a system in the MBL phase is prepared out of equilibrium, e.g. by changing one of the system parameters (quantum quench), there is no transport of particles or heat, and thus it is unable to thermalize, retaining a memory of the initial state and displaying a slow growth of quantum entanglement - properties that are of great interest for quantum information science. MBL provides a robust mechanism for the failure of thermalization in strongly disordered systems. Indeed, MBL is an extension of the well-known Anderson localization of single particles in the presence of disorder[8] to many-body, interacting systems. Theoretical progress in studies of thermalization and MBL systems[1–5] has been accompanied

by experimental advances and reports of MBL in (almost) isolated synthetic many-body systems with tunable interactions and disorder, such as ultracold atoms in optical lattices[9–14], trapped ions[15], superconducting qubits[16], and spins of nitrogen-vacancy centers in diamond[17]. However, a number of key questions remain open regarding the stability and the existence of the MBL phase. In particular, while long-range interactions may be expected to favor thermalization and short-range interactions with strong disorder may lead to MBL, it is not well established precisely how the range of interactions affects the dynamical behavior and the existence of MBL[18–27], and whether true MBL even exists in the limit of an infinite system, especially in $D > 1$. But even if the system does eventually thermalize, on shorter time scales it can exhibit the phenomenology of the true MBL phase[5]. The nature of the slow dynamics in such a prethermal or MBL-like regime is a subject of great interest and debate. Importantly, theoretical predictions can be tested only in a prethermal regime, since the true MBL does not exist in imperfectly isolated systems and, in any experiment, coupling to an external bath is unavoidable. For that reason, the experimentally relevant case of power-law interactions, which fall off as $\propto 1/r^\alpha$ with distance, in a $D$-dimensional system have been of special interest, but

[1]National High Magnetic Field Laboratory, Florida State University, Tallahassee, FL 32310, USA. [2]Department of Physics, Florida State University, Tallahassee, FL 32306, USA. [3]Department of Physics, Zhejiang Sci-Tech University, Hangzhou 310018, China. ✉e-mail: dragana@magnet.fsu.edu

also a challenge for theory[18–27]. Moreover, finite-size effects have often complicated the interpretation of both numerical simulations and experiments[4]. Therefore, there is a clear need for experiments in real, electronic materials with Coulomb interactions; because of their larger system sizes, such systems are much closer to the thermodynamic limit than synthetic ensembles of interacting, disordered particles. However, observing signatures of MBL in solid-state materials has been a challenge because the coupling between electrons and phonons makes it difficult to isolate the system from its thermal environment. Some evidence suggestive of MBL, in particular the apparent vanishing of the conductivity ($\sigma$) at a temperature $T > 0$, was reported in crystalline $In_2O_{3-x}$[28] and amorphous $In_xO$ films[29–31], but alternative interpretations have been also proposed[32]. Therefore, experimental studies of nonequilibrium dynamics on low-dimensional, $D > 1$ electronic materials are needed to probe the existence of a MBL-like, prethermal regime in strongly disordered systems with power-law interactions. This is precisely what our work provides by investigating quantum quench dynamics in $D = 2$ electron systems with $\alpha = 1$ and $\alpha = 3$ interactions, respectively.

We report the effect of the range of interactions on thermalization in the conductivity of a disordered 2D electron system (2DES) in Si metal-oxide-semiconductor field-effect transistors (MOSFETs), the basic building blocks of modern electronics, following a quantum quench protocol. This system is an excellent candidate for observing the MBL for the following reasons. First, total electron density $n_s$ can be varied easily by up to three orders of magnitude by changing the voltage $V_g$ on the metallic gate (Fig. 1a), thus allowing the study of thermalization dynamics across the quantum metal-insulator transition (MIT). At low $n_s$ of interest, the primary cause of the disorder in Si MOSFETs are charged impurities ($Na^+$) that are randomly distributed in the oxide and thus spatially separated from the 2DES; they are effectively frozen below 300 K (ref. 33). Furthermore, it is well known that the electron-phonon coupling between the 2DES and bulk Si, as well as in bulk Si itself, is very weak at low enough temperatures $T$ (refs. 34,35). In particular, for our devices in the relevant range of $n_s$, heat transfer between the 2DES and the environment is dominated by electron diffusion through the contacts (drain and source in Fig. 1a), as opposed to phonons, at $T \lesssim 1.6$ K (refs. 34,35). The heat transfer further proceeds through the metallic measurement leads attached to the contacts. To reduce thermal coupling to the environment even further, we placed the samples and leads in vacuum (Methods), unless noted otherwise. Finally, the MOSFET structure provides the option of screening the Coulomb interaction within the 2DES by reducing the thickness of the oxide ($d_{ox}$) that separates the 2DES from the gate (Fig. 1a). In particular, the metallic gate at a distance $d_{ox}$ from the 2DES creates an image charge for each electron, leading to the interaction law $\propto (1/r - 1/\sqrt{r^2 + 4d_{ox}^2})$, where the second term accounts for the Coulomb interaction between the electron's image charge and another electron in the 2DES[36–40]. At large distances $r \gg 2d_{ox}$, this potential falls off in a dipolelike fashion, as $\propto 1/r^3$. Therefore, at low enough densities such that the mean electron separation $2a = 2(\pi n_s)^{-1/2} \gg 2d_{ox}$, a condition more easily satisfied in devices with a smaller $d_{ox}$, electrons interact as though they were dipoles.

We focus on two sets of Si MOSFETs that were manufactured simultaneously, using identical procedure, the only difference being the value of $d_{ox}$ (Methods). In thick-oxide devices, $d_{ox} = 50$ nm, similar to most Si MOSFETs used in the studies of a 2D MIT[41,42]. In the density regime of interest, $5 \lesssim d_{ox}/a \lesssim 8$ and the Coulomb interaction is long-range, i.e. $\propto 1/r$. In thin-oxide devices, $d_{ox} = 6.9$ nm and $0.7 \lesssim d_{ox}/a < 1.5$, so that the screened Coulomb interaction is $\propto 1/r^3$. Hereafter we refer to it as the short-range interaction, and also note that here the screening by the gate is stronger than in other ground-plane screening studies in which the corresponding ratio $d/a$ ($d$ is the distance to the ground plane) was larger [e.g. $d/a \sim 1.5 - 2.5$ for $In_xO$ films[43]]. The 4.2 K peak mobility of the 2DES, a rough measure of the amount of

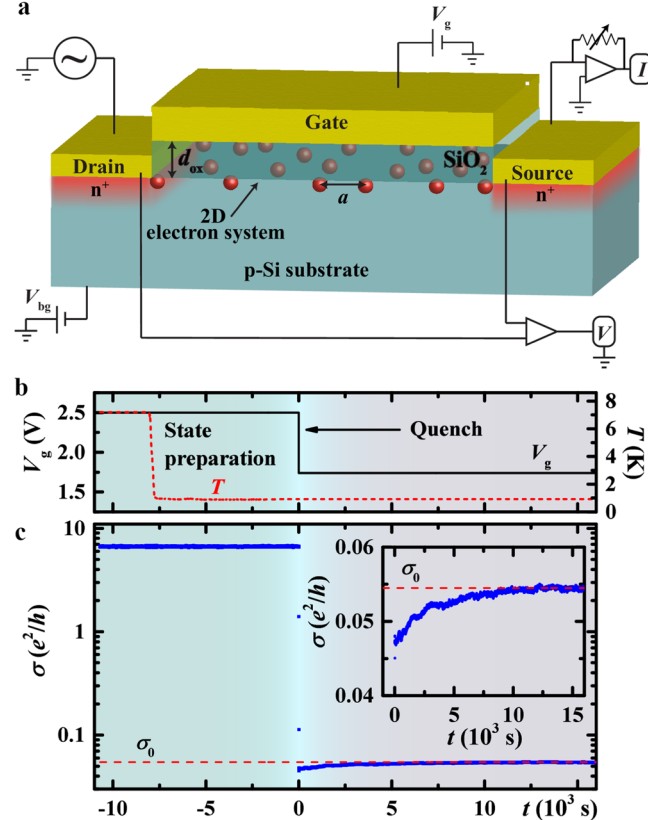

**Fig. 1 | Experimental investigation of quench dynamics in a 2D electron system. a** Schematic diagram of a Si MOSFET and the measurement set-up (Methods). Samples and the (metallic) measurement leads are mounted in vacuum. At low $T < 2$ K, the 2DES is connected to a thermal bath mainly through the measurement leads which are attached to the source and drain contacts. When the mean electron separation ($a$) is larger than the distance from the metallic gate ($d_{ox}$), the Coulomb interaction is modified from the long-range ~ $1/r$ to a screened or short-range ~ $1/r^3$ form. Total electron density $n_s$ is controlled by the gate voltage $V_g$. **b** $V_g$ and $T$ vs time ($t$) in a typical experimental protocol. Nonequilibrium dynamics is launched at $t = 0$ by a rapid change of $n_s$ from its initial to a final state value. **c** Conductivity $\sigma(t)$ corresponding to the protocol in **b** in the short-range case (sample 2 × 20): $V_g^i = 2.5$ V [$n_s^i (10^{11} cm^{-2}) = 32.2$], $V_g^f = 1.74$ V [$n_s^f (10^{11} cm^{-2}) = 8.44 > n_c$]; $T = 0.92$ K. Inset: Expanded view of the relaxation after $n_s$ change. Red dashed line indicates the (time-averaged) equilibrium conductivity in the final state, $\sigma_0(n_s^f, T)$. Source data are provided as a Source Data file.

disorder[33], indicates that our devices are relatively strongly disordered[44,45] (Supplementary Fig. 1). Previous studies have established that the equilibrium transport properties of the 2DESs in these two sets of high-disorder Si MOSFETs[44,45] are qualitatively the same, i.e. they are not affected by the range of the Coulomb interaction. For example, it was found that $\sigma(n_s, T)$ is essentially the same in both sets of devices (Supplementary Fig. 2), with the MIT occurring at similar critical densities, $n_c(10^{11} cm^{-2}) = 5.0 \pm 0.3$ and $n_c(10^{11} cm^{-2}) = 4.2 \pm 0.2$ for thick- and thin-oxide samples, respectively[45] (see also Supplementary Note 1 for more details on the MIT). In fact, even the critical exponents associated with the MIT are the same, consistent with a disorder-dominated nature of the MIT[45]. The onset of localization at $n_c$ implies that here the characteristic energy scale for disorder ($W$) becomes comparable to and exceeds the Fermi energy $E_F[K] = 7.31 n_s[10^{11} cm^{-2}]$ (ref. 33), i.e. $W \sim 35$ K. At the same time, the ratio $r_s$ of the average Coulomb energy per electron to the Fermi energy, $r_s = E_C/E_F \propto n_s^{-1/2} \sim 4$ (ref. 33). In thin-oxide devices, this ratio is reduced to $r_s \lesssim 1$ because of the screening by the gate[36], so that all three energy scales ($W$, $E_F$, and $E_C$) are comparable.

Here we demonstrate that, in contrast to equilibrium transport, there is a striking difference in the nonequilibrium dynamics of the 2DESs in these two sets of high-disorder Si MOSFETs, i.e. depending on the range of the Coulomb interaction. In the long-range case, charge dynamics near the 2D MIT was probed previously using a variety of experimental protocols that included studies of both relaxations of $\sigma$ with time ($t$) after applying a large perturbation and fluctuations of $\sigma(t)$ as a result of a small perturbation. The results have revealed the intrinsic, glassy behavior of the charge degrees of freedom, indicating that the 2D MIT is closely related to the melting of the Coulomb glass (see ref. 46 for a review; also Supplementary Note 2), consistent with theoretical expectations[47]. To explore the effect of screening the Coulomb interaction, we focus on the quantum quench protocol in which $n_s$ is changed rapidly by a large amount (relative to $E_F$). We find that, in the short-range case, the glassy dynamics does not persist; instead, we see negligible relaxation and strong sensitivity of the nonequilibrium behavior to thermal coupling to the environment, consistent with the proximity to the MBL phase. Therefore, our results reveal a transition from thermal to MBL-like dynamical behavior in a 2D system as the interaction range is reduced for a fixed disorder strength.

## Results

### Evolution of $\sigma$ with time after a rapid change of total electron density

Measurements are performed on a number of samples, which are labeled according to their dimensions $L$ [$\mu$m] $\times$ $W$ [$\mu$m] ($L$ - length, $W$ - width; see Methods). A typical experimental procedure (Fig. 1b) involves cooling the sample down from a high temperature (usually $7 - 20$ K) to a measurement $T$ with $V_g$ fixed at a value corresponding to a high initial carrier density, $n_s^i \gg n_c$. $\sigma(T)$ at such high $n_s^i$ is very weak (see, e.g., Supplementary Figs. 2 and 1c), as observed also in other Si MOSFETs with a large amount of disorder. The quench dynamics is then induced at time $t = 0$ by reducing $n_s$ rapidly (within 2 s) by a large amount to its final value, $n_s^f$, while monitoring the time evolution of the conductivity (Fig. 1c). Generally, a state prepared in this way ($k_B T \ll E_F < \Delta E_F$) is highly out-of-equilibrium, as confirmed by a very long time needed for $\sigma(n_s^f, T, t)$ to reach a (time-averaged) stationary value $\sigma_0(n_s^f, T)$ (Fig. 1c inset). By performing a subsequent warm-up to $\sim 7 - 20$ K, where no relaxations are observed, and then a cooldown to the same measurement $T$, we obtain the same value of $\sigma_0(n_s^f, T)$ (Supplementary Fig. 3). This shows that $\sigma(n_s^f, T, t)$ had indeed fully relaxed to a stationary value within the measurement time, suggesting that $\sigma_0(n_s^f, T)$ represents the equilibrium conductivity for the given $V_g^f$ and $T$, i.e. that the system thermalizes. However, other types of measurements beyond the scope of this work are needed to establish whether $\sigma_0$ corresponds to a true thermal equilibrium or some stationary state.

In the case of the long-range interaction, the relaxations initially overshoot $\sigma_0$ and $\sigma$ continues to move away from it with time; it is only at some later time that $\sigma$ starts to approach $\sigma_0$, thus giving rise to a minimum in $\sigma(t)$, as shown in Fig. 2a. [The overshooting of $\sigma_0$ manifests itself as a maximum, not a minimum, in $\sigma(t)$ when $n_s^i < n_s^f$ (refs. 48,49).] As $T$ is reduced, the relaxations become slower: the minimum shifts to lower values of $\sigma/\sigma_0$ and to longer times, until it disappears from the experimental time window at low enough $T$. A detailed study of the relaxations[48] has found that the approach to $\sigma_0$ [i.e. at times after the minimum in $\sigma(t)$] is exponential, such that the characteristic time diverges exponentially with decreasing $T$: $\tau_\sigma \propto \exp(-E_A/T)$, i.e. $\tau_\sigma \to \infty$ as $T \to 0$ (see also Supplementary Note 2). This means that, strictly speaking, the 2DES cannot thermalize only at $T=0$. On shorter time scales, before the minimum in $\sigma(t)$, the relaxations are nonexponential ($\sigma/\sigma_0 \propto t^{-\alpha} \exp[-(t/\tau_{lo})^\beta]$, where $\alpha < 0.4$ and $0.2 < \beta < 0.45$)[48], consistent with the existence of many metastable states and the so-called hierarchical pictures of glasses[50]. Indeed, extensive studies of charge

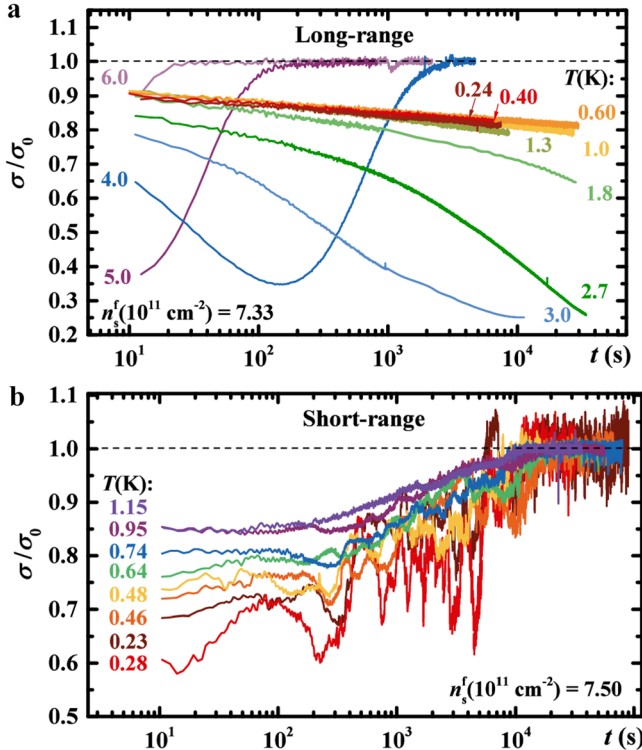

**Fig. 2 | Relaxations of conductivity $\sigma(t)$ normalized by the stationary value in the final state at a given temperature. a** Long-range case for $n_s^i (10^{11} \text{cm}^{-2}) = 20.26$ and $n_s^f (10^{11} \text{cm}^{-2}) = 7.33 \lesssim n_g$ at several $T$, as shown; sample $2 \times 50$. **b** Short-range case for $n_s^i (10^{11} \text{cm}^{-2}) = 32.20$ and $n_s^f (10^{11} \text{cm}^{-2}) = 7.50$; sample $2 \times 20$. Dashed black lines indicate the stationary value $\sigma = \sigma_0(n_s^f, T)$. In both **a** and **b**, $n_s^f > n_c$. Source data are provided as a Source Data file.

dynamics in these devices have established[42,44,48,49,51,52] glassy freezing as $T \to 0$ and glassy behavior at low enough $T$ and $t$ for all $n_s$ up to a glass transition density $n_g = (7.5 \pm 0.3) \times 10^{11}$ cm$^{-2}$, such that $n_c < n_g$. This gives rise to an intermediate phase ($n_c < n_s < n_g$) in which the dynamics is glassy, but the 2DES is a bad conductor ($k_F l < 1$, where $k_F$ is the Fermi wave vector and $l$ is the mean free path). These observations are consistent with theoretical expectations[47], with the Coulomb glass behavior ultimately resulting from the frustration induced by the competition of the long-range Coulomb interaction and disorder.

In the case of screened or short-range Coulomb interaction, we find both some similarities and important differences compared to the long-range case. This is illustrated in Fig. 2b, which shows $\sigma(t)$ measured at $n_s^f > n_c$ similar to the one in Fig. 2a. Here the deviations of $\sigma(t)$ from $\sigma_0$ also depend on $T$, and $\sigma(t)$ curves overshoot $\sigma_0$ before slowly returning to the apparent equilibrium value $\sigma_0$. However, in contrast to the long-range case, at shorter times the time evolution of $\sigma$ for a given $T$ is very weak and, surprisingly, $\tau_\sigma$ is similar for all $T$. Indeed, based on $\sigma(n_s^f, T, t)$ measured for different $n_s^f$ and $T$ (see also Supplementary Figs. 4 and 5 for representative data), we note the following key findings.

a) The slowly evolving $\sigma(n_s^f, T, t)$ are seen for $n_s^f (10^{11} \text{cm}^{-2}) \lesssim 17$ (Supplementary Fig. 4), which corresponds to $\sigma_0 < e^2/h$. In other words, the slow dynamics becomes observable on the metallic side of the MIT, but in the regime of strong disorder when $k_F l < 1$, in analogy with the long-range case. To explore the density dependence further, we plot the initial amplitude of the relaxations, defined as $\sigma(10 \text{ s})/\sigma_0$, at a fixed low $T$ (see Supplementary Fig. 6 for $T$ dependence at fixed $n_s^f$). Figure 3 shows that, as $n_s^f$ is reduced, these initial deviations from the apparent equilibrium become more pronounced and peak just before $n_c$ is reached, thus reflecting the presence of the underlying MIT. It is

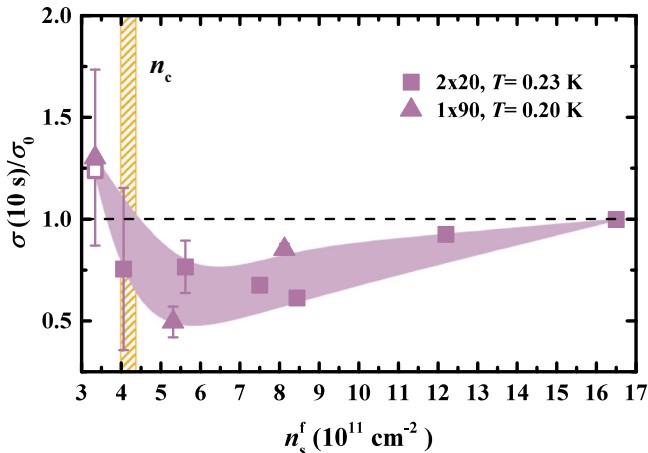

**Fig. 3 | Short-range case: the initial relaxation amplitude $\sigma(t=10\text{s})/\sigma_0$ vs final density $n_s^f$ at a low $T \approx 0.2$ K.** The relaxations become observable on the metallic side of the 2D metal-insulator transition (MIT), at $n_s^f(10^{11}\text{cm}^{-2}) \approx 17$ for which $\sigma_0 \sim e^2/h$, i.e. $k_F l < 1$. The relaxation amplitude increases as the density is reduced, and it peaks just before $n_c$, the critical density for the MIT, is reached. Symbol shapes indicate the size of the sample, as shown; open symbols describe the data obtained on another sample with the same dimensions. For all data, $n_s^i(10^{11}\text{cm}^{-2}) = (32.2 \pm 0.3)$. The vertical yellow hatched region shows $n_c$. Dashed black line corresponds to the apparent equilibrium value $\sigma = \sigma_0(n_s^f, T)$. The error bars reflect $\pm 1$ SD of the fluctuations of $\sigma_0$ with time. Source data are provided as a Source Data file.

interesting that a nonmonotonic density dependence of the relaxation amplitude near the MIT was observed also in the glassy dynamics of thick-oxide devices[51] (see also Supplementary Note 2).

Furthermore, we find that the noise increases as $n_s^f$ is reduced towards the MIT, especially at lower $T$ (Supplementary Fig. 4). Although similar behavior was observed in the long-range case[44,52], we note that the noise in the short-range case is much more pronounced, such that $\sigma(t)$ is dominated by the noise already for $n_s^f > n_c$ as the MIT is approached (Supplementary Fig. 4c). In the insulating regime, where transport occurs via 2D Mott variable-range hopping[45], large noise dominates $\sigma(t)$ (Supplementary Figs. 4d and 5) such that, in some cases, there are no visible relaxations. Indeed, the initial amplitudes $\sigma(10\text{ s})/\sigma_0$ become sample dependent (Supplementary Figs. 4d, 5, and 6). The observation of stronger noise in the short-range case is consistent with general expectations, since long-range interactions tend to suppress fluctuations. Studies of noise, i.e. fluctuations in $\sigma(t)$, provided important information about the nature of glassy dynamics and the free energy landscape in the 2DES with the long-range Coulomb interaction[44,52]. While studies of fluctuations with time have been suggested also as an alternative probe of the MBL dynamics[53], the noise analysis in the short-range case is beyond the scope of this work. This includes experimental protocols in which a small perturbation is applied to the system, such as a small change of density ($k_B T < \Delta E_F \ll E_F$) or temperature ($k_B T < k_B \Delta T \ll E_F$). In those cases, the measured signal is dominated by the noise and no visible relaxations are observed (see also Supplementary Note 2). We focus instead on the aspects of the relaxations observed in the current protocol that most starkly deviate from the long-range case.

b) In contrast to thick-oxide devices with time-dependent, glassy relaxations (Fig. 2a and ref. 48), in the case of screened Coulomb interaction the relaxations have a very weak time dependence at intermediate times (Fig. 2b and Supplementary Fig. 4). Moreover, the apparent thermalization time $\tau_\sigma$ seems to be independent of $T$ and $n_s^f$. Indeed, we find that the approach to $\sigma_0$ at long times may be fitted with an exponential function, $|\sigma - \sigma_0| \propto \exp[-(t/\tau_\sigma)]$, as illustrated in Supplementary Fig. 7 (there the data are shown on a log-linear scale, in contrast to linear-log plots in Fig. 2, Supplementary Figs. 4 and 5). This

allows us to extract $\tau_\sigma$ for all $n_s^f$ and $T$ where visible, exponential relaxations are observed at long times, and for different samples. The results of such fits indicate that the thermalization is anomalously slow, with $\tau_\sigma \sim 10^4$ s (Fig. 4a, solid symbols) and no systematic dependence on either $T$ or $n_s^f$. This is in a striking contrast to $\tau_\sigma \propto \exp(-E_A/T)$ in the long-range case[48] (Fig. 4a, open symbols; $E_A \approx 57$ K, independent of $n_s^f$), where the divergence of $\tau_\sigma$ was one of the signatures of glassy freezing as $T \to 0$. Therefore, in the 2DES with a screened Coulomb interaction, we find no evidence of glassy dynamics, which provides key insight into this fundamental problem[54]. What is the nature of the observed anomalously slow transport then?

## Sensitivity of the quench dynamics to coupling to the thermal bath

In contrast to glassy systems, the MBL phase is expected to be highly susceptible to the coupling to a thermal bath. In particular, while the thermalization time is expected to diverge for a completely isolated system, for even a small coupling to a thermal bath the system will eventually thermalize with the environment at very long times. At intermediate times, however, it will exhibit MBL properties[4,13]. The susceptibility of the system to external coupling can thus be used as an experimental signature of MBL[4,13]. Therefore, to test this scenario, we have performed some relaxation measurements with samples and measurement leads placed in $^4$He vapor ($T = 1.7$ K; Supplementary Fig. 8) to increase the thermal coupling to the environment (Methods). Figure 4b shows that, in that case, $\tau_\sigma$ is about an order of magnitude lower, i.e. the apparent thermalization is much faster, precisely as expected for MBL. This result indicates that the values of $\tau_\sigma$ are determined by the residual coupling of the 2DES to the outside world. Our measurements at $T \gtrsim 2$ K further confirm our conclusions.

At $T \gtrsim 2$ K, electron-phonon coupling between the 2DES and bulk Si increases the residual coupling to the environment[34,35], and $\tau_\sigma$ is reduced even further. To demonstrate this, first we note that a reasonably good estimate of $\tau_\sigma$ can be obtained also from $n_s$ sweeps. For example, for samples in vacuum at $T = 1.5$ K, where electron-phonon coupling is weak or negligible, even sweeps as long as $\sim 1.6 \times 10^4$ s result in a hysteresis in $\sigma(n_s)$ observed when $\sigma < e^2/h$ (Supplementary Fig. 9a). This gives a lower bound for $\tau_\sigma \sim 1.6 \times 10^4$ s, consistent with values obtained from relaxation measurements (Fig. 4a). At $T = 4.2$ K, though, there is no hysteresis regardless of whether the samples are placed in vacuum or immersed in liquid helium, and the same $\sigma(n_s)$ is obtained even for sweeps as fast as $\sim 200$ s (Supplementary Fig. 9b). This implies that $\tau_\sigma$ must be even lower, and that thermalization is dominated by electron-phonon coupling, as expected in this $T$ range[34,35].

We have thus shown that, in the short-range case, $\tau_\sigma$ increases by orders of magnitude as the residual coupling to the outside world is reduced: a) $\tau_\sigma < 200$ s (upper bound at $T = 4.2$ K) for strong coupling dominated by electron-phonon interactions at $T \gtrsim 2$ K; b) $\tau_\sigma \sim 10^3$ s for intermediate coupling due to $^4$He vapor at $T < 2$ K where electron-phonon interactions become weak or negligible; and c) $\tau_\sigma \sim 10^4$ s for weak coupling with samples in vacuum at $T < 2$ K. The observed behavior is, therefore, consistent with the MBL phase with nonzero coupling to a thermal bath, similar to synthetic many-body systems that exhibit MBL-like properties[4]. It is thus plausible that with any additional reduction of this residual coupling, if feasible, $\tau_\sigma$ would increase even further and likely diverge for a completely isolated system. In that case, it would always exceed the finite (for $T \neq 0$) thermalization time found in the long-range case, including at $T < 2$ K where thermalization with a long-range Coulomb interaction becomes immeasurably slow (Fig. 4a). While $\tau_\sigma$ in the short-range case is strongly reduced at $T \gtrsim 2$ K due to the increased electron-phonon coupling, in the long-range case, on the other hand, glassy dynamics persists even at $T \gtrsim 4$ K (Figs. 2a and 4a)[48]. This robustness of the glassy dynamics with respect to coupling to an external bath is indeed consistent with general expectations[4,55].

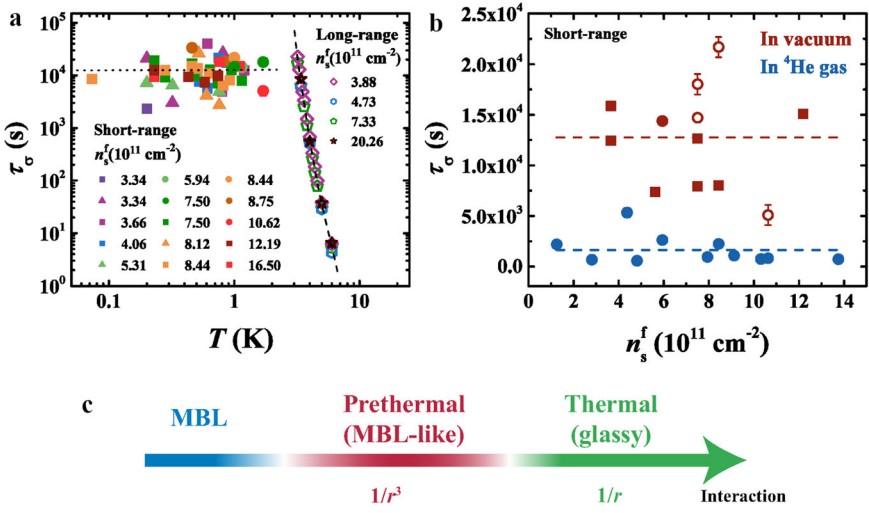

**Fig. 4 | Dependence of the thermalization time on the range of interactions and thermal coupling to the environment. a** Thermalization time $\tau_\sigma$ as a function of $T$ for different $n_s^f$, as shown. For the short-range case (solid symbols), the symbol shape indicates the sample (squares: $2 \times 20$, circles: $2 \times 50$, triangles: $1 \times 90$); $n_s^i(10^{11} cm^{-2}) = (32.2 \pm 0.3)$. Results for the long-range case (open symbols) are adapted from ref. 48 in which a $2 \times 50$ sample was studied for different $n_s^f$. Dashed black lines guide the eye. In both cases, the samples were placed in vacuum. Electron-phonon coupling is weak or negligible for $T < 2$ K, and it becomes dominant for $T > 2$ K. **b** $\tau_\sigma$ vs. $n_s$ for the short-range case with two different types of thermal coupling to the environment. The data from **a**, shown for $T = (0.9 - 1.7)$ K, obtained with samples placed in vacuum and thus very weakly coupled to a thermal bath, demonstrate that $\tau_\sigma$ is anomalously long. When measured in $^4$He gas at $T = 1.7$ K, the samples and measurement leads are more strongly coupled to the environment. In that case, $\tau_\sigma$ is about an order of magnitude lower, i.e. the thermalization is much faster. Open symbols are data from another sample with the same dimensions; dashed lines guide the eye. The error bars were determined from the fits at long times, as illustrated in Supplementary Fig. 7. **c** Schematic of the effect of the Coulomb interaction range on quench dynamics in a disordered 2DES at low enough $n_s$. In the long-range, $\sim 1/r$ case, the dynamics is glassy, but the system thermalizes, in principle at a finite $\tau_\sigma(T)$ at all $T > 0$ ($\tau_\sigma \to \infty$ as $T \to 0$). In the short-range, $\sim 1/r^3$ case, $\tau_\sigma$ is independent of $T$, but it is extremely long when coupling to the environment is very weak; when this coupling increases, $\tau_\sigma$ decreases. Therefore, in this case, the thermalization results from the residual coupling of the 2DES to the outside world, but on time scales short enough compared to $\tau_\sigma$ the system exhibits MBL-like properties. For **a** and **b**, source data are provided as a Source Data file.

## Discussion

Our findings are summarized schematically in Fig. 4c. The quench dynamics of a disordered, nearly thermally isolated 2DES with a screened Coulomb interaction ($\propto 1/r^3$) has revealed negligible, non-glassy relaxations of conductivity at intermediate times. At long times, we observe an approach back to an apparent equilibrium state with extremely long thermalization times ($\tau_\sigma \sim 10^4$ s) that are independent of $T$. The MBL-like nature of the observed slow dynamics has been confirmed by verifying that residual coupling to the environment sets the time scale for thermalization. In particular, by increasing the residual coupling, $\tau_\sigma$ is reduced by orders of magnitude. In case of the long-range Coulomb interaction ($\propto 1/r$), however, the MBL does not survive: the system thermalizes, although $\tau_\sigma(T)$ can be long because the dynamics is glassy. We note that, in both cases, slow dynamics is observed only when $k_F l < 1$, i.e. for strong enough disorder when the 2DES becomes a bad conductor. Since the two sets of devices were fabricated under identical conditions, any extrinsic effects, such as possible charging of the Si-SiO$_2$ interface traps, can be ruled out as the origin of the observed differences in their nonequilibrium behavior (see also Supplementary Note 2). Therefore, we have determined that the interaction range has a striking effect on the dynamics, although the equilibrium behavior of the 2DES is not affected.

Studies of atoms in 2D optical lattices[11,14] reported evidence of MBL-like dynamics by tracking the evolution of the local density with time. In contrast, yet complementing previous studies, we have detected MBL-like dynamics of charge transport by a direct measurement of the conductivity. In general, in a conducting system in the proximity to the MBL phase, i.e. in a prethermal regime, both $\sigma$ and the electron density should take a long time to thermalize. However, the local electron density in Si MOSFETs cannot be measured, while the total density $n_s$ is calculated from the gate voltage $V_g$ and the known device characteristics, in particular the oxide capacitance[33] (see Methods). When $V_g$ is changed, the total density $n_s$ has to change within the time constant of the device, $\tau = RC$, where $R$ is the resistance of the 2DES and $C$ is the total capacitance of the oxide. We estimate that the longest charging time in our study, corresponding to the largest sample resistance, is $\tau \sim 5$ ns (see Methods). In the experiment, $V_g$ is reduced within 2 s, which is the measurement resolution. The observation of an "instantaneous", orders-of-magnitude drop of the average $\sigma$ at $t = 0$ in Fig. 1c is indeed consistent with such a rapid change of the total or average $n_s$ (see also Supplementary Fig. 2). Clearly, the total density cannot change further with time because $V_g$ remains constant following the quench. However, this does not imply that electrons are thermalized: local density rearrangements can continue until a much longer time, at least until $\tau_\sigma$, which should give rise to the fluctuations of conductivity, as seen in the data (Fig. 2b, Supplementary Figs. 4 and 5). These fluctuations are obviously non-Gaussian, indicating that the system is not in equilibrium; the study of the noise will be a subject of future work.

By investigating $D = 2$ electron systems with power-law interactions $\propto 1/r^\alpha$, we have observed MBL-like dynamics for $\alpha = 3$ consistent with $D < \alpha < 2D$. On the other hand, for $\alpha = 1$ consistent with $\alpha < D$, we find that the system thermalizes, in agreement with theoretical expectations (e.g., see ref. 5 for a review). Although the possibility of many-body localization and MBL-like behavior in systems with power-law interactions has been explored in many theoretical studies, the case with $D < \alpha < 2D$ has been under debate (see, e.g., refs. 18–27). Therefore, our observation of MBL-like behavior in this regime provides important insights and constraints for the theory.

Our central results are thus the direct observation of the MBL-like, prethermal regime in an electronic system, and clarifying the effects of the interaction range on the fate of glassy dynamics and MBL in 2D.

However, whether the state reached at long times (at $t > \tau_\sigma$) in the case of a screened Coulomb interaction corresponds to a true thermal equilibrium or some quasi-steady state remains an open question for future study. Indeed, by establishing a new, versatile solid-state platform for the study of MBL, our work also opens new possibilities for further investigations, such as noise measurements as a probe of ergodicity breaking and many-body entanglement[4,53].

## Methods

### Samples

Our study was performed on two sets of rectangular $n$-channel (100)-Si MOSFET devices fabricated simultaneously using the 0.25-$\mu$m Si technology, with only a difference in the thickness of the oxide, $d_{ox}$. The samples have poly-Si gates, self-aligned ion-implanted contacts, substrate doping $N_a \sim 2 \times 10^{17}$ cm$^{-3}$, and oxide charge $N_{ox} \approx (1-1.5) \times 10^{11}$ cm$^{-2}$. Both the thick-oxide ($d_{ox} = 50$ nm) and thin-oxide ($d_{ox} = 6.9$ nm) MOSFETs were studied previously and described in more detail in refs. 44,48,49,51,52 and ref. 45, respectively. We present results from a thick-oxide sample with dimensions $L \times W$ ($L$ - length, i.e. source-to-drain distance, and $W$- width) of $2\,\mu m \times 50\,\mu m$, and seven thin-oxide devices with dimensions of $2\,\mu m \times 20\,\mu m$, $2\,\mu m \times 50\,\mu m$, and $1\,\mu m \times 90\,\mu m$. Large aspect ratios $W/L$ enable measurements of lower conductivities $\sigma = G/(W/L)$ ($G$ is the conductance), crucial for probing the insulating regime at the lowest electron densities $n_s$. Furthermore, wider samples have lower contact resistances $R_c \propto 1/W$ (ref. 56), with $R_c \sim$ several $\Omega$ in our devices[44,56]. Therefore, $R_c$ is always negligible relative to the sample resistance. Finally, the observation of single-parameter scaling near the MIT[45], in the $n_s$ and $T$ regimes of interest, confirms the absence of finite-size effects. In other words, the sample length $L > \xi$, where $\xi$ is the correlation length.

In analogy with previous studies[44,45,48,49,51,52] on these MOSFETs, all measurements were conducted with a back-gate (substrate) bias of $V_{bg} = -2$ V to ensure that all electrons in the inversion layer at the Si/SiO$_2$ interface populate a ground subband[33], i.e. that the system is 2D and with no scattering by local magnetic moments due to the population of the upper subbands. The total electron density $n_s$ was varied by the gate voltage $V_g$ such that $n_s = C_{ox}(V_g - V_{th})/e$, where $C_{ox}$ is the geometric capacitance of the oxide, $e$ is the charge of an electron, and $V_{th}$ is the threshold voltage[33] (see also Supplementary Note 3). For thick- and thin-oxide MOSFETs, respectively, $n_s(10^{11}\text{cm}^{-2}) = 4.31(V_g[V] - 6.3)$ and $n_s(10^{11}\text{cm}^{-2}) = 31.25(V_g[V] - V_{th})$, $V_{th}(V) = (1.47 \pm 0.01)$, with the corresponding total oxide capacitances $C_{thick} \approx 7 \times 10^{-17}$ F and $C_{thin} \approx 5 \times 10^{-16}$ F. In the regime of interest, the resistances $R$ of the 2DES are typically $R \sim 10^3 - 10^7\,\Omega$, so the charging time constant of the thin-oxide device, for example, is $\tau \sim 0.5$ ps $- 5$ ns. These values are consistent with the literature[56], and they are orders-of-magnitude smaller than our measurement resolution.

### Measurements

The standard two-probe ac lock-in method (typically at $\sim 11$ Hz) was used for conductivity measurements with an ITHACO 1211 current preamplifier, SR 5113 voltage preamplifier, and SR 7265 lock-in amplifiers. Two precision dc voltage standards (EDC MV116J) were used to apply the gate voltage $V_g$ and the back-gate bias $V_{bg}$. The excitation voltage was constant and low enough (typically 10 $\mu$V) to ensure that the conduction was Ohmic. The experiment was performed in a dilution refrigerator (0.05 K $< T < 1$ K) and a $^3$He system (0.24 K $< T < 200$ K), in which samples are mounted on a Cu cold finger in vacuum. Both cryostats have heavily filtered wiring that includes a 1 k$\Omega$ resistor in series with a $\pi$ filter [5 dB (60 dB) EMI reduction at 10 MHz (1 GHz)] in each wire at the room temperature end of the cryostat to reduce heating and phase decoherence by EM radiation. In addition, low-pass $RC$ filters with $R = 1$ k$\Omega$ and $C = 10$ nF are installed on each wire at the

mixing chamber stage of the dilution refrigerator. Some measurements were performed in a variable-temperature insert (1.5 K $< T < 200$ K), in which samples are placed in a $^4$He vapor, as well as in a liquid helium storage dewar with samples immersed in liquid helium ($T = 4.2$ K). Indeed, immersion of electrical leads in $^4$He is commonly employed in low-temperature set-ups to increase the thermal coupling to the environment, which occurs mainly via phonon transmission between the metallic leads and helium[57].

## Data availability

All data that support the findings of this study are included in the article and its Supplementary Information, and are available from the corresponding author upon request. Source data for the figures in the main text are provided with this paper. Source data are provided with this paper.

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

## Acknowledgements

We acknowledge helpful discussions with B. L. Altshuler, A. D. Mirlin, A. Polkovnikov, and L. Rademaker. This work was supported by NSF Grants Nos. DMR-1307075 (D. P.), DMR-1707785 (D. P.), DMR-2104193 (D. P.), and the National High Magnetic Field Laboratory (NHMFL) through the NSF Cooperative Agreements Nos. DMR-1157490, DMR-1644779, and the State of Florida.

## Author contributions

Si MOSFETs were designed by D.P. and fabricated in the Silicon Facility at IBM Thomas J. Watson Research Center; L.J.S., J.J., and P.V.L. performed the measurements; L.J.S and P.V.L. analyzed the data; L.J.S and D.P. wrote the manuscript, with input from all authors; D.P. supervised the project.

## Competing interests

The authors declare no competing interests.
