## [Peer Review File · Nature Communications]

I. REPORT

In this paper, the authors study the effect of range of interactions on conductivity of a Si MOSFET. Authors claim that MOSFET is a good system to explore many-body localization (MBL) because at low electron density, the main source of disorder are charged impurities which are frozen at low temperatures giving rise to a quenched random disorder. Further at low temperature, e-phonon coupling between 2d electrons and bulk Si is very weak for $T \leq 1.6K$ which help in realizing a system which is weakly coupled to the bath. Authors perform a quantum quench by changing the gate voltage and measure the time evolution of the charge conductivity. It is definitely an interesting study to explore the experimental realization of MBL and prethermal MBL phase in real materials which is really the need for the community, there are a couple of technical issues due to which I can not recommend this manuscript in the current form for publication in Nature Communications.

- Authors claim that the interactions are Coulomb type with $1/r$ form for the thickness of oxide = $50nm$ and for the reduced thickness of $6.9nm$ it is of dipolar form $1/r^3$. Though it is true that for a thinner oxide there should be a better screening of the e-e interaction due to the gate, but having a quantitative estimate about the range of interaction based on the oxide thickness is a bit strong claim. It would be nice to have some evidence in support of this very quantitative statement about the range of interactions.
- Authors basically perform a quantum quench by changing the gate voltage and study the time evolution of the conductivity. It is reported in the manuscript that for the “long range case” the e density changes by a factor of 3 from $20.26 \cdot 10^{11}cm^{-2}$ to $7.33 \cdot 10^{11}cm^{-2}$ in 2 sec by changing the gate voltage and for the “short range case” initial density of $32.20 \cdot 10^{11}cm^{-2}$ changes to $7.5 \cdot 10^{11}cm^{-2}$ again in 2 sec. How these e densities have been measured? How do authors conclude that density changes by a large factor of 3 or 4 in 2 sec? It would be appreciated if a plot of time evolution of e density vs time can be added to the manuscript to explain this.
- Further, the observation shows that though the e density is changed within 2 secs by a large amount, system takes a very long time more than (10000sec) to reach a stationary value of conductivity in the “short range” system. This slow dynamics has

been interpreted as a signature of precursor to MBL phase, namely the “prethermal MBL like phase”. If the system under consideration is really close to being in the MBL like phase, then not only the conductivity but also the e density should take a large time to thermalize. Thus, I find the two observations, namely, e density changing by a large factor in just 2 sec and conductivity reaching a saturation value in more than 10000sec very counter intuitive. An explanation in this context would make the manuscript more comprehensive.

- At the end, I would like to mention that there is a large amount of work done on long-range interacting MBL systems which provide evidence in favour of an MBL like phase even in the presence of long range interactions provided the interactions do not change sign and the hopping remains short enough in range. In the experimental situation considered, it is only the e-e interactions which are getting screened by changing the thickness of the oxide and the range of hopping remains unchanged, authors should make an attempt at least to analyse the experimental data along the light of these works as well and make a statement about it in the manuscript. It is even more important because the time scale at which conductivity saturates even in “long range” system at low temperatures is comparable to the “short range case” as shown in Fig 4. I have mentioned here some of these works which are probably missed by the authors Phys. Rev. B 93, 245427 (2016); PRB 91, 094202 (2015); PRB 92, 104428 (2015); Phys. Rev. Lett. 113, 243002 (2014); Phys. Rev. B 99, 224203 (2019); SciPost Phys. 7, 042 (2019).

To conclude, this manuscript is an interesting attempt to explore the MBL phase in real materials but the manuscripts require revision. Clarifications to the above mentioned points would be appreciated.

REVIEWER COMMENTS

Reviewer #2 (Remarks to the Author):

Stanley et al. closely studied the relaxation process of conductivity after a fast change of gate voltage and investigated the non-equilibrium process. They discussed different effects of the Coulomb interaction range in a disordered two-dimensional electron system at adequately low carrier density. In particular, there are crossover from many body localization states to glassy states. The study is quite thorough and the interpretation is pretty convincing.

In terms of solidity of this study, I have no question publishing it. However, this study might not attract a wide attention in its current form. I recommend the authors to significantly revise the abstract and introduction part to make it accessible to wider audience to fit Nature communications as a quite comprehensive journal. Importantly, it should be useful to highlight the impact of the results in a larger picture.

Reply to the Reviewers' Reports

We are grateful to the reviewers for reading our manuscript and for their constructive comments that have improved the quality of our paper. Our response to the reviewers' comments is given below (comments in blue, our response in black).

Reviewer #1 (Remarks to the Author):

In this paper, the authors study the effect of range of interactions on conductivity of a Si MOSFET. Authors claim that MOSFET is a good system to explore many-body localization (MBL) because at low electron density, the main source of disorder are charged impurities which are frozen at low temperatures giving rise to a quenched random disorder. Further at low temperature, e-phonon coupling between 2d electrons and bulk Si is very weak for $T \leq 1.6\text{K}$ which help in realizing a system which is weakly coupled to the bath. Authors perform a quantum quench by changing the gate voltage and measure the time evolution of the charge conductivity. It is definitely an interesting study to explore the experimental realization of MBL and prethermal MBL phase in real materials which is really the need for the community, there are a couple of technical issues due to which I can not recommend this manuscript in the current form for publication in Nature Communications.

We are grateful to the reviewer for recognizing the significance and broad interest of our study. We have addressed the technical issues raised by the referee as described below.

- Authors claim that the interactions are Coulomb type with $1/r$ form for the thickness of oxide = 50nm and for the reduced thickness of 6.9nm it is of dipolar form $1/r^3$. Though it is true that for a thinner oxide there should be a better screening of the e-e interaction due to the gate, but having a quantitative estimate about the range of interaction based on the oxide thickness is a bit strong claim. It would be nice to have some evidence in support of this very quantitative statement about the range of interactions.

We thank the referee for prompting us to provide further clarification on this issue. In the previous version of our manuscript, we had omitted this information for brevity, as it is well-known from the earlier literature on gate screening (e.g., see ref. 45 and refs. therein, as well as refs. 36-40). Nevertheless, we agree with the reviewer that a more detailed explanation is needed for a broader audience. Therefore, we have added the following text at the end of paragraph 2 of the Introduction. We note that the references 37- 40 are new inclusions in this revised version of the manuscript.

“In particular, the metallic gate at a distance d_{ox} from the 2DES creates an image charge for each electron, leading to the interaction law $\propto (1/r - 1/\sqrt{r^2 + 4d_{\text{ox}}^2})$, where the second term accounts for the Coulomb interaction between the electron's image charge and another electron in the

2DES³⁶⁻⁴⁰. At large distances $r \gg 2d_{\text{ox}}$, this potential falls off in a dipolelike fashion, as $\propto 1/r^3$. Therefore, at low enough densities such that the mean electron separation $2a=2(\pi n_s)^{-1/2} \gg 2d_{\text{ox}}$, a condition more easily satisfied in devices with a smaller d_{ox} , electrons interact as though they were dipoles.”

Furthermore, in paragraph 3 of the Introduction, we have made the corresponding, stylistic changes to the text. The values $0.7 \lesssim d_{\text{ox}}/a < 1.5$ for thin-oxide MOSFETs had been already provided earlier, so that the complete information now clarifies “that the screened Coulomb interaction is $\propto 1/r^3$ ”.

In this version, we have added similar information about thick-oxide MOSFETs for comparison and completeness, as follows.

“In the density regime of interest, $5 \lesssim d_{\text{ox}}/a \lesssim 8$ and the Coulomb interaction is long-range, i.e. $\propto 1/r$.”

- Authors basically perform a quantum quench by changing the gate voltage and study the time evolution of the conductivity. It is reported in the manuscript that for the “long range case” the e density changes by a factor of 3 from $20.26 \cdot 10^{11} \text{cm}^{-2}$ to $7.33 \cdot 10^{11} \text{cm}^{-2}$ in 2 sec by changing the gate voltage and for the “short range case” initial density of $32.20 \cdot 10^{11} \text{cm}^{-2}$ changes to $7.5 \cdot 10^{11} \text{cm}^{-2}$ again in 2 sec. How these e densities have been measured? How do authors conclude that density changes by a large factor of 3 or 4 in 2 sec? It would be appreciated if a plot of time evolution of e density vs time can be added to the manuscript to explain this.

This question is related to the next one, so we provide a comprehensive answer to both questions below the reviewer’s next comment, including how we have modified our manuscript to address them. Here we just note that the electron density in Si MOSFETs is not measured directly, but rather the **total or average** density is calculated using the known device characteristics. We have added the corresponding formula and text in Methods, specifically at the end of the “Samples” subsection, as follows.

“... $n_s=C_{\text{ox}}(V_g - V_{\text{th}})/e$, where C_{ox} is the geometric capacitance of the oxide, e is the charge of an electron, and V_{th} is the threshold voltage³³.”

- Further, the observation shows that though the e density is changed within 2 secs by a large amount, system takes a very long time more than (10000sec) to reach a stationary value of conductivity in the “short range” system. This slow dynamics has been interpreted as a signature of precursor to MBL phase, namely the “prethermal MBL like phase”. If the system under consideration is really close to being in the MBL like phase, then not only the conductivity but also the e density should take a large time to thermalize. Thus, I find the two observations, namely, e density changing by a large factor in just 2 sec and conductivity reaching a saturation value in more than 10000sec very counter intuitive. An explanation in this context would make the

manuscript more comprehensive.

We thank the reviewer for bringing up this important point, which indeed required further clarification. Therefore, we have added an entire paragraph, paragraph 2 in the Discussion, to provide a more detailed explanation, as follows. Additionally, reference 14 is a new inclusion in this revised version.

“We note that we have detected MBL-like dynamics of charge transport by a direct measurement of the conductivity, in contrast to earlier studies, such as those on atoms in 2D optical lattices^{11,14}, which tracked the evolution of the density with time. In our conducting system in the proximity to the MBL phase, i.e. in a prethermal regime, both σ and the electron density n_s should indeed take a long time to thermalize. The electron density in Si MOSFETs cannot be measured directly; it is calculated instead from the gate voltage V_g and the known device characteristics, in particular the oxide capacitance³³ (see Methods). When V_g is changed, the total density n_s has to change within the time constant of the device and the circuit, $\tau=RC$, where R and C are the equivalent resistance and capacitance of the entire measurement set-up, respectively. We estimate that the longest charging time in our study, corresponding to the largest sample resistance, is $\tau\sim 10$ ms. In the experiment, V_g is reduced within 2 s, i.e. within the measurement resolution. The observation of an “instantaneous”, orders-of-magnitude drop of the average σ at $t=0$ in Fig. 1C is indeed consistent with such a rapid change of the total or average n_s (see also Supplementary Fig. 2). However, this does not imply that electrons are thermalized: although the total (average) density changes quickly, local density rearrangements can continue until a much longer time, at least until τ_σ , which should give rise to the fluctuations of conductivity, as seen in the data (Fig. 2B, Supplementary Figs. 4 and 5).”

Therefore, the answer to the reviewer’s previous question is that the average density has changed within ~ 2 s, as indicated by the behavior of the average σ at time $t=0$ shown in Fig. 1C, consistent with the known strong gate-voltage dependence of the average σ at a fixed T (Suppl. Fig. 2). We believe that the above new paragraph makes it clear that all the experimental observations are consistent with each other and with the system being in the prethermal regime in the case of a screened, dipolar Coulomb interaction.

- At the end, I would like to mention that there is a large amount of work done on long-range interacting MBL systems which provide evidence in favour of an MBL like phase even in the presence of long range interactions provided the interactions do not change sign and the hopping remains short enough in range. In the experimental situation considered, it is only the e-e interactions which are getting screened by changing the thickness of the oxide and the range of hopping remains unchanged, authors should make an attempt at least to analyse the experimental data along the light of these works as well and make a statement about it in the manuscript. It is even more important because the time scale at which conductivity saturates even in “long range” system at low temperatures is comparable to the “short range case” as shown in Fig 4. I have mentioned here some of these works which are probably missed by the authors Phys. Rev. B 93, 245427 (2016); PRB 91, 094202 (2015); PRB 92, 104428 (2015); Phys. Rev. Lett. 113, 243002 (2014); Phys. Rev. B 99, 224203 (2019); SciPost Phys.

7, 042 (2019).

We thank the reviewer for drawing our attention to these papers. The theoretical literature on this topic is indeed considerable, as there is no apparent consensus on the precise role of the interaction range. Even the terminology “long-range” vs “short-range” does not seem to be consistent across the literature. Therefore, from the experimental point of view, it is difficult to make strong statements about the applicability of various theoretical models to our experimental situation. Nevertheless, we have made several changes to our manuscript to make our statements more precise and more consistent with the existing literature, as well as to highlight their relevance to theoretical studies on this topic. We have also expanded the list of references by citing the papers mentioned by the reviewer: they now appear as refs. 18-21, 25, 26.

In particular, to clarify the precise range of interactions studied in our experiment, we now state explicitly throughout the paper, starting from the revised abstract, that we study the case of power-law interactions $\propto 1/r^\alpha$, where we consider the situations with $\alpha=1$ and $\alpha=3$. Also, in paragraph 3 of the Introduction, we have added the following sentence about the screened, dipolar Coulomb interaction.

“Hereafter we refer to it as the short-range interaction, and also note that here the...”

To address the reviewer’s comment, we have added an entire paragraph (paragraph 3) in the Discussion section of our paper, as follows.

“By investigating $D=2$ electron systems with power-law interactions $\propto 1/r^\alpha$, we have observed MBL-like dynamics for $\alpha=3$ consistent with $D < \alpha < 2D$. On the other hand, for $\alpha=1$ consistent with $\alpha < D$, we find that the system thermalizes, in agreement with theoretical expectations (e.g., see ref.⁵ for a review). Although the possibility of many-body localization and MBL-like behavior in systems with power-law interactions has been explored in many theoretical studies, the case with $D < \alpha < 2D$ has been under debate (see, e.g., refs.¹⁸⁻²⁷). Therefore, our observation of MBL-like behavior in this regime provides important insights and constraints for the theory.”

Finally, to emphasize the observed differences between the two cases even further, as related to τ_σ , we have made the following changes.

a) In paragraph 1 of the Discussion, we have added the following sentence to emphasize that the value of τ_σ is strongly dependent on the residual coupling when $\alpha=3$.

“In particular, by increasing the residual coupling, τ_σ is reduced by orders of magnitude.”

b) On the other hand, the robustness of the glassy dynamics with respect to coupling to an external bath was already emphasized in the sentence that just precedes the Discussion. In this revised version, we have added reference 55.

To conclude, this manuscript is an interesting attempt to explore the MBL phase in real materials but the manuscripts require revision. Clarifications to the above

mentioned points would be appreciated.

We thank the reviewer for the valuable questions and suggestions that have helped us to improve our manuscript. We hope that the reviewer will find the revised version of our paper suitable for publication in Nature Communications.

Reviewer #2 (Remarks to the Author):

Stanley et al. closely studied the relaxation process of conductivity after a fast change of gate voltage and investigated the non-equilibrium process. They discussed different effects of the Coulomb interaction range in a disordered two-dimensional electron system at adequately low carrier density. In particular, there are crossover from many body localization states to glassy states. The study is quite thorough and the interpretation is pretty convincing.

In terms of solidity of this study, I have no question publishing it. However, this study might not attract a wide attention in its current form. I recommend the authors to significantly revise the abstract and introduction part to make it accessible to wider audience to fit Nature communications as a quite comprehensive journal. Importantly, it should be useful to highlight the impact of the results in a larger picture.

We thank the reviewer for appreciating the high quality and importance of our study. We are also grateful to the reviewer for suggestions on how to make our manuscript more accessible to a wider audience of Nature Communications.

Therefore, following the reviewer's suggestions, we have *completely* revised the abstract and the first, introductory paragraph of our manuscript, as shown using track changes/color highlighting. The abstract and the introduction are now much more general and understandable to a broader audience, like the literature on the same or similar topics published in high-profile journals such as Nature Communications.

We have also less focused less on the details of our samples. For example, instead of discussing long-range and screened Coulomb interactions in the abstract, introduction, and discussion, we specify that we consider electronic systems in two dimensions ($D=2$) with power-law interactions $\propto 1/r^\alpha$, where $\alpha=1$ and $\alpha=3$, respectively. This is more understandable to a general audience, and it makes the relevance of our study clearer; please see also our response to the comments of reviewer #1 related to the comparison of our results to theory. Moreover, since our results provide important information on the possibility of MBL-like behavior in these cases that have been a subject of debate in the literature (see also new paragraph 3 in the Discussion), these revisions also highlight the impact of our results in a larger context, as suggested by the reviewer.

In addition, to highlight the impact of our results even further, in the introduction we have noted that "finite-size effects have often complicated the interpretation of both numerical simulations and experiments⁴. Therefore, there is a clear need for experiments in real, electronic materials with Coulomb interactions; because of their larger system sizes, such systems are much closer to

the thermodynamic limit than synthetic ensembles of interacting, disordered particles” that were studied previously. The revised abstract also emphasizes this aspect of our study and its differences from prior experiments in this field.

We thank the reviewer again for the valuable suggestions that have helped us to improve our manuscript and make it more accessible to a wider audience. Thus we hope that the reviewer will find the revised version of our paper suitable for publication in Nature Communications.

REVIEWER COMMENTS

Reviewer #1 (Remarks to the Author):

Authors have replied to some of the comments made in the first report and also modified the presentation of the manuscript accordingly. But the main question about the time dependence of the e-density has not been answered. Authors simply mentioned the relation used to estimate the average e-density.

But as I wrote in my last report as well "If the system under consideration is really close to being in the

MBL like phase, then not only the conductivity but also the e density should take a large time to thermalize."

Thus, I think authors must try to use Hall measurement and obtain time dependence of e- density as a consistency check on the slow dynamics claimed.

Secondly, I still don't see a clear difference in the time scales of dynamics in what authors call "short-range" and "long-range". In Fig 4A, If one does not look at large temperature data, even for "long-range" case the time scales are comparable to what is described as "short-range".

The main experimental observation is that the Si MOSFET system under study has very slow dynamics for certain parameter regime. Of course MBL like prethermal regime can be one possible explanation for this, but how do authors rule out other possibilities like that of a glassy phase throughout the parameter range explored?

Once authors take care of the above mentioned points, I will be happy to recommend it for publication. But in the absence of Hall measurement and time dependent e-density data and also the absence of an explanation to rule out other possible slow dynamic phases like glass, it looks like a weak case to me.

Reviewer #2 (Remarks to the Author):

The authors has revised the manuscript based on my suggestions and now the quality of the manuscript is significantly improved. I recommend publishing it after careful proofreading.

Reply to the Reviewer's Report

We thank the reviewer for providing a second report on our manuscript, and for the comments that have helped us to improve our paper further. Our response to the reviewer's comments is given below (comments in blue, our response in black).

Reviewer #1 (Remarks to the Author):

Authors have replied to some of the comments made in the first report and also modified the presentation of the manuscript accordingly. But the main question about the time dependence of the e-density has not been answered. Authors simply mentioned the relation used to estimate the average e-density.

We kindly request the reviewer to consider that we had also included a detailed explanation in our previous response. Additionally, we had made significant revisions to the Discussion section of our manuscript, adding a new, extended paragraph that addresses the topic mentioned (bottom of p. 16 and top of p. 17 in the previous version). To ensure clarity and eliminate any potential misunderstanding, we have provided an even more comprehensive response below, delving into greater detail regarding the matter at hand.

But as I wrote in my last report as well "If the system under consideration is really close to being in the MBL like phase, then not only the conductivity but also the e density should take a large time to thermalize."

Thus, I think authors must try to use Hall measurement and obtain time dependence of e- density as a consistency check on the slow dynamics claimed.

We agree with the reviewer that, indeed, both the conductivity (diffusivity) and the local electron density should take a long time to thermalize in the prethermal regime. This is why we had modified the previous version of our paper accordingly, that is, by adding such a sentence on p. 16 within the new paragraph mentioned above.

To answer the reviewer's question about the time dependence of the electron density in more detail and explain the related changes in the revised manuscript, we first note the following important points.

a) The 2D electron system (2DES) that we study forms one plate of a capacitor, as shown in Fig. 1a. The second plate of that capacitor is a metallic gate. The 2DES is adjacent to two heavily doped (n^+) regions (source and drain in Fig. 1a), which act like reservoirs of electrons that can be exchanged freely with the 2D electron layer. In other words, n^+ regions are connected to the 2DES - they represent contacts. When voltage V_g applied to the gate is changed, the **total (average) electron density** on the other capacitor plate, i.e., in the 2D layer, must change because the 2D electron layer is capacitively coupled to the gate via the electric field in the oxide; hence the name **field-effect** transistor (see refs. 33, 56). The capacitance is dominated by

the geometry and permittivity of the oxide, so the total density is given by $n_s = C_{\text{ox}}(V_g - V_{\text{th}})/e$ (see Methods). At low temperatures, such as those used in our study, when V_g is changed, electrons from the 2DES go into or out of contacts (n^+ regions) depending on the direction of V_g change. (A more precise model for a Si MOSFET is that of a distributed resistance-capacitance network, but that does not change these considerations.)

Therefore, when V_g is changed, the **total density** n_s (charge on the other capacitor plate) **has to change within the time constant of the device**, $\tau = RC$, where R is the resistance of the 2DES and C is the total capacitance of the oxide. In the regime of interest, $R \sim 10^3 - 10^7 \Omega$. For our thin-oxide MOSFETs, for example, $C \sim 5 \times 10^{-16} \text{ F}$, so that $\tau \sim \mathbf{0.5 \text{ ps} - 5 \text{ ns}}$. These values are consistent with the literature (e.g., ref. 56) and they are orders-of-magnitude smaller than our measurement resolution, as we noted previously (see Discussion and our previous reply). This means that the change of the total (average) density in our experiment seems “instantaneous”.

b) In our experiment, this initial, “instantaneous” reduction of the total (average) density represents the quench, i.e., the preparation of the state far from equilibrium. This is analogous to the initial removal of atoms from one side of the system (e.g., right side) in experiments on ultracold atoms in optical lattices, such as the one in ref. 11, reporting MBL in 2D. In that experiment (ref. 11), they then tracked the time evolution of the density imbalance (left vs right) in the system, which was possible because their experimental technique is a **local** measurement.

The reviewer’s question is about what happens with the electron density in our study following the quench, i.e., after the total density has been reduced rapidly, within the measurement resolution ($\sim 2 \text{ s}$). Clearly, the **total density cannot change further with time** because V_g remains constant and our system is a capacitor (see above). However, this does **not** imply that electrons are thermalized: **local density rearrangements** can continue until a much longer time, at least until τ_σ . The conductivity of the 2DES is affected by both the average n_s , which results from the uniform potential (the field-effect of the MOS capacitor), and by a random component that results from disorder and a nonuniform electron density. Therefore, local density rearrangements should give rise to the fluctuations of conductivity, as seen in the data (Fig. 2b, Supplementary Figs. 4 and 5). These fluctuations are obviously non-Gaussian, indicating that the system is not in equilibrium, but the study of noise in the short-range case is beyond the scope of this manuscript and it will be a subject of future work. We note, though, that such studies have been performed on the 2DES with the long-range Coulomb interaction [as mentioned at the bottom of p. 11 of the previous version of our paper; see refs. 44, 52, also Jaroszynski et al., Phys. Rev. Lett. **89**, 276401 (2002), Jaroszynski et al., Phys. Rev. Lett. **92**, 226403 (2004)]. Those studies provided evidence consistent with collective rearrangements of electrons with a hierarchical free energy landscape characteristic of a glass.

Since there are no local measurement techniques for Si MOSFETs (they have been a challenge to develop for technical reasons, i.e., mainly due to the presence of a metallic gate), it is not possible to perform local, spatially resolved measurements of the electron density with time following the quench. It seems that perhaps this is what the reviewer was interested in, possibly inspired by experiments on ultracold atoms. However, we **can** track the conductivity of the 2D system with time, in contrast to experiments on nonconducting ultracold atoms that can track

only the density evolution (e.g., refs. 11, 14). This is another novel, **complementary** contribution of our work to the literature on MBL.

c) The reviewer suggests the use of **Hall measurements** to obtain electron density. However, the well-known expression for the Hall resistivity, $R_{xy} = B/(ne)$, where n is the carrier density and B is the magnetic field, has been derived within the Drude theory of metallic conduction, i.e., it is valid when $k_{Fl} > 1$. **Our results**, in contrast, are observed in the $k_{Fl} < 1$ regime, where the 2DES is a bad conductor, i.e., near the metal-insulator transition (e.g., Fig. 2) and in the insulating regime. The Drude expression does **not** hold for $k_{Fl} < 1$. The behavior of the Hall effect as the MIT is approached, where $k_{Fl} < 1$, has been the subject of many experimental and theoretical studies some time ago [e.g., see experiments in Dai et al., Phys. Rev. Lett. **70**, 1968 (1993), Dai et al., Phys. Rev. B **49**, 14039 (1994), Teizer et al., Phys. Rev. B **67**, 121102(R) (2003), Breznay et al., Proc. Natl. Acad. Sci. U.S.A. **113**, 280–285 (2016), Gerber et al. Phys. Rev. B **95**, 214206 (2017), and references therein], but the question about the effects of disorder and localization on the Hall effect remains unresolved. In particular, there is no consensus in the literature on what Hall effect measures as the MIT is approached ($k_{Fl} < 1$). Since an adequate theory for that regime is still lacking, it is **unclear how to relate Hall measurements to electron density**.

Even for $k_{Fl} > 1$, the Drude formula is valid for a steady state. However, in our study we are interested in what happens **far from equilibrium**, following the quench, but as we noted above, the Hall effect in the $k_{Fl} < 1$ regime is not well understood even in the equilibrium case. Therefore, although Hall measurements at first glance might seem like an attractive option to provide information about electron density in our experiment, any interpretation of the data in the regime of interest would be unclear, unsubstantiated, and unreliable.

Finally, we note that this issue is even more complex because magnetic fields can affect equilibrium transport and dynamics of a 2DES in different ways, as demonstrated by the huge body of literature in this field. So far, we have not studied the effects of a magnetic field on any properties of a 2DES in thin-oxide MOSFETs, i.e., in the case of a short-range (dipolar) Coulomb interaction. Based on the extensive studies of Si MOSFETs with a long-range Coulomb interaction by both our group and several others (e.g., see review in ref. 46), we expect that the critical density n_c for the MIT will increase with field so that, for a given n_s , the 2DES will be even deeper in the insulating or poorly conducting regime than in zero field. This is why the behavior of $n_c(B)$ would be one of the things that would need to be established before any attempts to measure Hall resistivity. Furthermore, the studies on thick-oxide Si MOSFETs near the MIT were performed with magnetic fields parallel to the 2DES to avoid orbital effects and obtain information on the role of spin degrees of freedom. Hall measurements, of course, require fields perpendicular to the 2DES, where orbital effects are also important. Therefore, comprehensive **studies of equilibrium transport properties in both parallel and perpendicular magnetic fields** need to be completed **first** on these devices. Within that context, it would be interesting to perform Hall measurements near the MIT to add to the literature on this topic and to see whether the results would provide some insight into this long-standing, open problem. However, as mentioned above, any interpretation would remain speculative in the absence of further theoretical progress in this area. It is only after these equilibrium studies are complete that studies of the dynamics for a given B should be pursued, in

analogy with prior work on 2DESs with long-range Coulomb interactions. Clearly, the studies outlined here would constitute an ambitious, multi-year research program.

In response to the reviewer's comment, we have made the following changes in the manuscript based on the above points; we note that all the changes are shown in detail in the resubmitted version using highlighting.

i) We have modified paragraph 2 of the Discussion to make the distinction between the total density and local density clearer. (We also corrected the values of $\tau=RC$ and C – the previous text was a typo, but that does not affect anything else in the paper.) Paragraph 2 of the Discussion now starts with the following.

“Studies of atoms in 2D optical lattices^{11,14} reported evidence of MBL-like dynamics by tracking the evolution of the local density with time. In contrast, yet complementing previous studies ...”

The same paragraph ends with the following new sentence.

“These fluctuations are obviously non-Gaussian, indicating that the system is not in equilibrium; the study of the noise will be a subject of future work.”

ii) We have inserted “total” (electron density) in several places throughout the text, such as the heading of the first subsection under Results and Fig. 1a caption.

iii) In Methods (under “Samples”), we have added information about $\tau=RC$. We have also added Supplementary Note 3 with the following text, including the new Supplementary reference 18.

“For completeness, we mention that in Si MOSFETs at high carrier densities, such that $k_F l > 1$, electron density can be determined also from transport measurements in perpendicular magnetic fields (B). The carrier density obtained from low-temperature Hall measurements at relatively low fields ($\omega_c \tau_s < 1$, where ω_c is the cyclotron frequency and τ_s is the scattering time) is typically¹⁴ somewhat different from $n_s = C_{ox}(V_g - V_{th})/e$. It may also depend on the values of B and T used in the Hall measurement¹⁸ because of the quantum corrections to the conductivity, in particular, electron-electron interactions in the presence of disorder. However, at low carrier densities in the $k_F l < 1$ regime, which is the subject of our study, the Hall effect is not well understood, and thus it remains unclear how to relate Hall measurements to electron density.”

Secondly, I still don't see a clear difference in the time scales of dynamics in what authors call "short-range" and "long-range". In Fig 4A, If one does not look at large temperature data, even for "long-range" case the time scales are comparable to what is described as "short-range".

The reviewer did not state explicitly why this seems to be a concern, so we can only try to guess what exactly the reviewer had in mind and point out the following.

In the long-range, glassy case, the thermalization time $\tau_\sigma \propto \exp(-E_A/T)$. On the other hand, in the short-range case τ_σ is a temperature-independent constant (for a given thermal coupling to the environment). Hence, there **must** be a temperature at which the thermalization time in the glassy case is comparable to τ_σ in the short-range case. This is, indeed, seen in Fig. 4a, as the reviewer noticed. Furthermore, as discussed in more detail in the last paragraph of the subsection “Sensitivity...” (in Results), in the short-range case τ_σ depends practically exponentially on the thermal coupling to the bath so that τ_σ would diverge if this coupling could be reduced to zero.

We are also guessing that the reviewer believes that we draw conclusions about the nature of the dynamics based on the long thermalization time τ_σ found in the short-range case ($\sim 10^4$ s in Fig. 4a). We emphasize that this is certainly **NOT** true, as explained in more detail in response to the reviewer’s next comment. We agree with the reviewer (if this is what they had in mind) that this kind of reasoning would be flawed. This is illustrated best with an example of a simple resistor-capacitor (RC) circuit, in which the (dis)charging of the capacitor is an exponential process in time, with the characteristic (dis)charging time given by RC . Obviously, this time can be made arbitrarily long with a suitable choice of R and C , but such a long time would definitely not imply a glassy or MBL-like dynamics! We draw our conclusions instead based on other properties of the observed dynamics, as discussed in the manuscript and below in response to the reviewer’s next comments.

The main experimental observation is that the Si MOSFET system under study has very slow dynamics for certain parameter regime.

This is only partially true. As we remarked above, slow relaxation *per se* would not be sufficient to draw any conclusions about the nature of the dynamics. Our main experimental observations instead are **negligible relaxations** of conductivity at intermediate times and thermalization time that is **highly susceptible to the coupling to a thermal bath**. Both properties are precisely what is expected in the case of MBL-like dynamics. In fact, it is the sensitivity to thermal coupling to the environment that has been proposed as a key experimental signature of MBL behavior (see refs. 4, 13) and a way to distinguish it from glassy dynamics. This is why we devoted a separate subsection (“Sensitivity of the ...”) under Results to this topic.

To remove a possible misunderstanding and emphasize our main experimental findings, we have made the following changes in the manuscript.

- i) In the last paragraph of the Introduction, just before Results, the modified text now reads “... instead, we see negligible relaxation and strong sensitivity of the nonequilibrium behavior to thermal coupling to the environment...”.
- ii) The heading of the second subsection of the Results now reads “Sensitivity of the quench dynamics...” instead of the previous “Sensitivity of the relaxations...” to avoid a possible misunderstanding of the word “relaxations”.
- iii) In the first paragraph of the Discussion, we make a clear distinction between observations at intermediate times and those at long times: “...negligible, non-glassy relaxations of conductivity

at intermediate times. At long times, we observe an approach back to an apparent equilibrium state with extremely long thermalization times...”.

Of course MBL like prethermal regime can be one possible explanation for this, but how do authors rule out other possibilities like that of a glassy phase throughout the parameter range explored?

There are several reasons why the dynamics observed in the short-range case cannot be attributed to glassiness.

First, the manifestations of glassiness are nearly universal in a large class of both 3D and 2D systems that are out of equilibrium (e.g., spin glasses, supercooled liquids, granular films). In other words, their phenomenology is very similar even though the microscopic mechanisms leading to glassy dynamics may be very different. For example, our extensive studies of the dynamics in the long-range case (see reviews in refs. 42 and 46) demonstrated that glassy dynamics of the electrons in a 2DES is very similar to the behavior of 3D spin glasses. The underlying reason for such similar phenomenology is that glasses are characterized by a “rugged” free energy landscape, consisting of many metastable states separated by energy barriers with a broad distribution of heights. On the other hand, the slow dynamics that we report in the short-range case does not exhibit **any** properties that are characteristic of glasses, and the dynamics is strikingly different from that in the long-range case.

In particular, there is a huge, qualitative difference in the time dependence of the conductivity at intermediate times between the long-range case with glassy dynamics (Fig. 2a) and the short-range case with MBL-like dynamics (Fig. 2b) – that difference is evident from Fig. 2. As we discussed above, in the short-range case the relaxations are negligible, i.e., they do not exhibit any apparent time dependence at intermediate times. On the other hand, in the long-range, glassy case, the relaxations at intermediate times are nonexponential, obeying $\sigma/\sigma_0 \propto t^{-\alpha} \exp[-(t/\tau_{l0})^\beta]$ with $\alpha < 0.4$, $0.2 < \beta < 0.45$ (ref. 48), similar to spin glasses. Both power-law and stretched exponential relaxations are considered typical signatures of glassy behavior and reflect the existence of a broad distribution of relaxation times. There is **NO** such relaxation observed in the short-range case.

In addition, there is a difference in the temperature dependence of the thermalization time τ_σ in the glassy and MBL-like cases that was already discussed above. In the long-range case, $\tau_\sigma \propto \exp(-E_A/T)$, so that there is a temperature ($T=0$) at which τ_σ diverges – this is the glass transition temperature. In the short-range case, τ_σ is independent of temperature and determined only by the coupling to a bath.

Indeed, it has been suggested (see ref. 4) that the key way to distinguish an MBL system and a glass experimentally is to study the sensitivity of their dynamics to the coupling to a thermal bath: while glasses are robust with respect to coupling to an external bath, MBL dynamics is extremely sensitive to and suppressed by such coupling. This is discussed at the beginning of our second subsection (“Sensitivity...”) under Results. Our experimental observations, which are described in that same subsection, are consistent with these expectations. The robustness of

the glassy dynamics with respect to coupling to an external bath is discussed in the last couple of sentences of that same subsection.

Finally, our results exhibit precisely what is expected in the case of MBL-like dynamics, as we discussed above and as we stressed even further in the revised manuscript. There are no other, known slow dynamic phases.

To summarize, the slow dynamics that we report in the short-range case does not exhibit **any** properties that are characteristic of glasses, and it is strikingly different from that in the long-range case. The latter had been investigated in detail in a series of papers cited in our manuscript, and glassy dynamics is well-established there. Figure 2 shows that the dynamics in the short-range case is obviously of a different type and, furthermore, it agrees with the expectations for the MBL-like, prethermal regime.

To emphasize the differences between glassy dynamics and our observations in a short-range case further, we have made the following changes in the manuscript.

i) We have added the expression “ $(\sigma/\sigma_0 \propto t^{-\alpha} \exp[-(t/\tau_{l0})^\beta]$ where $\alpha < 0.4$ and $0.2 < \beta < 0.45$)” in the description of nonexponential relaxations in Fig. 2a (paragraph 2 in Results).

ii) We have added “with time-dependent, glassy relaxations” in paragraph 6 in Results. This serves to juxtapose the behavior at intermediate times in the long-range and short-range cases.

Once authors take care of the above mentioned points, I will be happy to recommend it for publication. But in the absence of Hall measurement and time dependent e-density data and also the absence of an explanation to rule out other possible slow dynamic phases like glass, it looks like a weak case to me.

We thank the reviewer for the additional comments, which we have addressed in detail both above and in the revised manuscript. They have helped us to improve our paper further.

In particular, we have emphasized that our results were obtained in the $k_F l < 1$ regime, in which it is not known how to relate Hall measurements to electron density; other, local density measurements are not feasible in Si MOSFETs. Furthermore, we have provided ample comparisons between the well-established glassy dynamics in the long-range case, characteristic of a broad class of glassy materials, and the new results observed in the short-range case – they clearly and obviously rule out glassy dynamics in the latter. The observed behavior of the conductivity agrees instead with the predictions for the MBL-like, prethermal regime, while there are no other, known slow dynamic phases.

Since we have addressed the reviewer’s comments in the revised manuscript, we hope the reviewer finds it suitable for publication in Nature Communications.

REVIEWERS' COMMENTS

Reviewer #1 (Remarks to the Author):

The authors have addressed all the queries and comments in detail and have also revised the manuscript based on my comments.

I appreciate the detailed response that authors have prepared in response to comments on thermalization of electron density, the difference in dynamics for long and short-range systems as well as about glassy dynamics. Though I still think it would be interesting to do Hall measurement on this system.

Overall the quality of the manuscript is significantly improved. The manuscript addresses a very important issue in the field of condensed matter physics and the experimental results are interesting. I recommend publishing this work in Nature Communications.